# END-TO-END STORY PLOT GENERATOR

## ABSTRACT

Story plots, while short, carry most of the essential information of a full story that may contain tens of thousands of words. We study the problem of automatic generation of story plots, which includes story premise, character descriptions, plot outlines, etc. To generate a single engaging plot, existing plot generators (e.g., DOC (Yang et al., 2022a)) require hundreds to thousands of calls to LLMs (e.g., OpenAI API) in the planning stage of the story plot, which is costly and takes at least several minutes. Moreover, the hard-wired nature of the method makes the pipeline non-differentiable, blocking fast specialization and personalization of the plot generator. In this paper, we propose three models, `OpenPlot`, `E2EPlot` and `RLPlot`, to address these challenges. `OpenPlot` replaces expensive OpenAI API calls with LLaMA2 (Touvron et al., 2023) calls via careful prompt designs, which leads to inexpensive generation of high-quality training datasets of story plots. We then train an end-to-end story plot generator, `E2EPlot`, by supervised fine-tuning (SFT) using approximately 13000 story plots generated by `OpenPlot`. `E2EPlot` generates story plots of comparable quality to `OpenPlot`, and is $> 10\times$ faster (1k tokens in only 30 seconds on average). Finally, we obtain `RLPlot` that is further fine-tuned with RLHF on several different reward models for different aspects of story quality, which yields 60.0% winning rate against `E2EPlot` along the aspect of suspense and surprise.

## 1 INTRODUCTIONS

Storytelling has been of great importance throughout human history. In ancient eras, narratives served as an important medium for disseminating knowledge between generations. Even today, the significance of stories remains undiminished since they continue to shape our cultural paradigms, foster empathy, and convey various ideas. Beyond its historical importance, people love reading stories because they get to experience events that might never happen to them.

In the era of large language models (LLMs), readers have greater access to stories as they can now be crafted not only by human authors but also by LLMs. Many previous works can automatically generate short stories, where the length ranges from several sentences to a couple of paragraphs (Fan et al., 2018; Yao et al., 2019; Rashkin et al., 2020; Han et al., 2022). However, even a short story from a human perspective typically consists of thousands of words, which is a long context for a language model. It has long been a challenge for AI to generate long-form, coherent stories (Charniak, 1972; Turner, 2014), let alone interesting ones. While there have been advances in dealing with long context for LLMs (Chen et al., 2023a; Han et al., 2023; Chen et al., 2023b; Xiao et al., 2023), on a broader scale, generating high-quality long outputs remains a major challenge even for the most advanced LLMs such as GPT4 (OpenAI, 2023) and Llama 2 (Touvron et al., 2023).

A more recent line of work (Yang et al., 2022b;a; Zhou et al., 2023) focuses on improving the quality of generated stories in terms of length and coherence. By generating stories hierarchically, calling LLMs recursively, or adding more detailed control during the generation, these methods are able to generate longer and higher-quality stories than a rolling-window baseline, i.e., iteratively prompting LLMs such as ChatGPT to continue writing. However, these previous methods consist of complex hard-wired procedures. For example, the DOC method (Yang et al., 2022a) generates hierarchical outlines with one bullet point per LLM call in a breadth-first order. To ensure the quality of each bullet point, DOC requires multiple responses per prompt and performs rejection sampling. This requires *thousands of calls* to InstructGPT3-175B (text-davinci-002) in the planning stage that generates story plots, which is costly and time-consuming compared to the plain rolling-window

method and is hard to improve by fine-tuning since the generation procedure is hard-wired and thus not differentiable. Thus, it is appealing to have an *end-to-end* model, which not only has fast generation speed and can be further fine-tuned but can also generate high-quality stories comparable to the state-of-the-art method.

However, it might be overly ambitious to aim for an end-to-end model that can directly generate a complete story that is long and of high quality, even with SoTA LLMs. Therefore we focus on story *plot* generation, which has two benefits. First, the story plots are much shorter than stories in their entire forms, are highly structured and carry almost all the essential information of the story. This makes it much easier for LLMs to learn and generate, for reward models to give preference signals, and for humans to read and evaluate. Second, story plots can lead to detailed story rendering via existing pipelines (e.g., DOC (Yang et al., 2022a)) by calling LLMs with proper prompting and logit controls (Yang & Klein, 2021), which makes them convenient bridges that connect the entire story.

In this work, we aim to create an end-to-end story plot generator that can automatically generate story plots with one LLM forward pass. To train such an end-to-end model, we first re-implement the story plot generation procedure, namely `OpenPlot`, of the DOC method using Llama2-13B-chat (Touvron et al., 2023). `OpenPlot` is a fully reproducible and parallelizable pipeline, compared to the DOC pipeline which relies on OpenAI APIs and may subject to model update and rate limit that are out of reach by end users.

Using `OpenPlot`, we generate a large batch of story plots, which are used to fine-tune an end-to-end model, namely `E2EPlot`, based on Llama2-7B-chat. `E2EPlot` is able to generate story plots of comparable quality to `OpenPlot`, according to GPT4 (OpenAI, 2023) evaluation results among overall qualities (see Section 3 for details), and at a 10x faster speed (about 1000 tokens in 30 seconds), while the generation speed of DOC is much slower, taking several minutes to generate a story plot by calling OpenAI APIs many times.

Finally, to demonstrate that `E2EPlot` is capable of specializing into specific human feedback, we further fine-tune it along various aspects related to story quality (e.g., interestingness, coherence, good ending, etc.). Specifically, we train several different reward models, one for each aspects according to GPT4 (OpenAI, 2023) evaluation results (see Section 3 for details), and further fine-tune `E2EPlot` by RLHF to improve the quality of the generator. The resulting model, `RLPlot`, yields 60% win rate against `E2EPlot` along the aspect of suspense and surprise.

## 2    METHODS

In this section, we introduce our full pipeline for building the end-to-end story plot generator (`OpenPlot`, `E2EPlot` and `RLPlot`) in details. In Section 2.1, we re-design the existing DOC pipeline (Yang et al., 2022a) by Llama2 to get rid of the restriction of rate limits and unpredictable model update in calling OpenAI APIs. The resulting pipeline, `OpenPlot`, enables large batch generation, allowing us to create a large training dataset. In Section 2.2, we use the training dataset to fine-tune the Llama2-7B-chat model (Touvron et al., 2023) to obtain an end-to-end model, `E2EPlot`. In Section 2.3, we further train several reward models for different aspects and improve `E2EPlot` using RLHF to yield `RLPlot`.

### 2.1    PIPELINE FOR CREATING DATASETS: OPENPLOT

We first discuss the motivation for modifying the previous pipeline and the major challenges in Section 2.1.1 and then discuss the solutions in detail in Section 2.1.2. More details of the exact format of the prompts we used are deferred to Appendix A.

### 2.1.1    MOTIVATION AND MAJOR CHALLENGES

To train an end-to-end story plot generator, we first need to create a large training dataset consisting of thousands of story plots. The previous DOC pipeline (Yang et al., 2022a) is a promising candidate for creating a high-quality training set since it is one of the state-of-the-art approaches for story plot generation. However, it cannot be directly applied to our case for the following reasons. First, the original DOC method heavily relies on OpenAI API calls, and thus the generation procedure is largely constrained by the OpenAI API rate limit if done in parallel. Second, it uses a completion

model that supports suffix (i.e., a model that can do text infilling given a text prefix and a text suffix, Bavarian et al. (2022)) to perform most tasks, while many advanced LLMs, such as GPT4 (OpenAI, 2023) and Llama2 (Touvron et al., 2023), are chat models and do not support suffix.

To this end, we follow the logic of Yang et al. (2022a) to build the pipeline while replacing text-davinci-002 with Llama2-13B-chat (Touvron et al., 2023). However, replacing text-davinci-002 with Llama2-13B-chat introduces new challenges for the story plot generation. Below, we first list three major challenges and the high-level ideas of corresponding solutions, and then discuss the solutions in detail in Section 2.1.2.

- **Challenge 1: How to generate the outline in a breadth-first and coarse-to-fine manner and leverage proper contextual information?** The story plot generated by the DOC pipeline contains the Premise, Setting, Characters, and Outline (see Table 7 for the exact form of story plots). The outline is hierarchical and contains two levels of bullet points[1]. The DOC method generates one bullet point at a time in breadth-first order (or, equivalently, a coarse-to-fine manner), i.e., it first generates all the top-level bullet points and then expands each top-level point (e.g., Point 1) by generating sub-level points (e.g., Point 1a, Point 1b, etc.) under it. This coarse-to-fine manner is consistent with the way humans make plans. However, note that the second-level points need to be consistent with not only the previous and current top-level outlines but also the subsequent top-level outlines. For example, the bullet point 2a needs to be consistent with Point 1 and Point 2, but also needs to be consistent with the content of Point 3 since it has already been generated. Therefore, one should include as many existing outline points as possible regardless of the relative position. An ideal solution is that if we have access to a completion model which supports suffix, we can put the preceding points in the prompt and the subsequent points in the suffix, which is helpful to keep the consistency of the whole outline.

- **Challenge 2: How to use chat models to substitute for completion models that support suffix?** The previous DOC method with text-davinci-002, a completion model that supports suffix, adopted the above method to generate the hierarchical outline. However, since we use Llama2 for our rebuilt pipeline and Llama2 is a chat model that does not accept a suffix, we must develop a new solution to keep the consistency of the generated outline. We address this issue by simulating a completion model using a chat model. The high-level idea is that we start the prompt with detailed instructions on how to perform a completion task, then provide the suffix before the original prompt. By our observation, it would be better to provide the suffix first and then the original prompt, since the generated content usually continues with the end of the whole prompt. More details are discussed in Section 2.1.2 and Appendix A.

- **Challenge 3: How to maintain the quality of the story?** The DOC pipeline generates one point at a time and requires hundreds of LLM calls, and a single failed step could derail the whole story. To prevent this, the DOC pipeline generates multiple candidate responses and selects the best one at each step. However, due to the performance difference between different LLMs, after replacing text-davinci-002 with Llama2-13b-chat, the prompt for the original DOC pipeline does not necessarily work for our rebuilt pipeline. Therefore, we carefully designed the prompts with detailed instructions to ensure the quality of the generated story. We will discuss some concrete examples of the instructions in the prompt in Section 2.1.2.

### 2.1.2 OVERVIEW AND DISCUSSION OF THE PIPELINE

We discussed the high-level solution to major challenges for our rebuilt DOC pipeline, `OpenPlot`, in Section 2.1.1. In this section, we discuss more details of the prompt design for the pipeline. One can refer to Appendix A for the exact form of prompts we use in the pipeline, and examples of the structure of story plots are provided in Table 7.

**Premise.** The first step is to generate a premise for the story. For the previous DOC pipeline, it suffices to provide the prompt of "Write a premise for a short story." For `OpenPlot`, the same

---

[1]Actually, the DOC method supports different numbers of levels for the hierarchical outline. In this paper, we focus on a two-level hierarchy.

prompt might result in a response with an unexpected format. To ensure that the format of the output is structured and thus easy to process, we end the prompt with "Premise: " to enforce that the language model's response starts after "Premise: " [2]

**Setting.** After obtaining the premise of the whole story, the next step is to infer the story's setting from the premise. This step is relatively simple and does not require additional design strategies.

**Characters.** One of the most important elements in a story is the characters, whose intrinsic motivations and interactions with each other are vital to the trajectory of the whole story. Based on the premise and setting, we generate characters one by one. For each character, we first generate their full name and then their portrait.

**Remark 2.1.** *In the original DOC method, one only needs to end the prompt with "Character Portrait: " to guide the LLM to generate the portrait. However, when we replace the text-davinci-002 engine with the Llama2 model, due to the performance difference between these two models, the Llama2 model tends to output a longer description for the portrait. Moreover, the output of Llama2 focuses more on the age and appearance of the character instead of occupation, experiences, or relationships with other characters (e.g., "Tom is 22 years old and has brown curly hair"). Therefore, we add detailed instructions on generating portraits such as "focusing on relationship between characters, occupation and experience instead of appearance" to ensure the generated portraits contain more useful information about the character.*

After generating the first character, we can repeat the above procedure to generate more characters. We can control the number of major characters in the story, and in our implementation, we set the desired number to be 3-6, since too few characters is likely to make the story boring, and too many characters may reduce opportunities for characters to interact with each other.

**Outline.** After generating all the major characters, we are ready to build the main skeleton of the story. Similar to DOC, we aim to create a hierarchical outline. In this paper, we create two-level hierarchical outlines, where the top level typically contains four bullet points, and the second level contains three to four subpoints under each top-level point. We also generate the outline in breadth-first order, i.e., we first generate the top-level outline (numbered as 1, 2, 3, . . .) and then generate the sub-level outline (numbered as 1a, 1b, . . ., 2a, . . .).

**Remark 2.2.** *To make the generated story plots more reasonable, we add corresponding instructions. First, to control the length of the top-level outline, we require the LLM to use no more than 4 points. Second, to make the generation procedure more stable, we require the LLM to generate only one point at a time. Third, during the preliminary experiments, we found that the generated plots sometimes have a missing ending; to address this issue, we explicitly ask the LLM to make sure that the generated top-level outline has a clear ending. A tricky point is that adding "IMPORTANT: Please" significantly improves the quality of the generated outline regarding a clear ending.*

After generating the top bullet point 1, we can include the content of point 1 in the prompt and continue to generate the subsequent points until the whole top-level outline is complete. Since we generate the outline in breadth-first order, we will expand each top-level node in sequence. Note that it is important to keep the generated subpoint consistent with not only the previous points but also the subsequent points. To achieve this, the original DOC method uses a completion model (text-davinci-002) and adds a suffix containing the content of all subsequent points. Since we use Llama2 in our rebuilt pipeline, which is a chat model and thus does not accept suffixes, we need to simulate a completion model using a chat model.

**Remark 2.3.** *To simulate a completion model using a chat model, we need to include all the contents in the prompt for the chat model and add detailed instructions. First, we observe that putting the content of the original prompt after the suffix makes it easier for the LLM to continue with the original prompt. Second, we also need to add instructions at the beginning of the whole prompt, such as "Your output should not contain the content of the suffix. Only use the suffix as complementary information" to obtain desired results since otherwise, the LLM will memorize the context of the suffix, and the output will be nearly identical to the suffix. There is also a detailed instruction in the prefix (i.e., the original prompt for the completion model) on how to generate the current sub-level*

---

[2]Note that this is a guiding prompt strategy (Saravia, 2022).

*points, such as "generating one or two points without repeating the content of the suffix and stop", which can "reinforce" the requirement that the output should not repeat the content of the suffix. See more details and the exact format of the prompt and instructions in Appendix A.*

After expanding each top-level point by generating sub-points, we obtain a hierarchical outline, and the story plot generation procedure is finished.

## 2.2 TRAINING END-TO-END MODEL: E2EPLOT

After building the pipeline for batch generation of story plots, we can use the generated dataset to train an end-to-end model. We first generated 16000 story plots and then filtered them by excluding bad-structured outlines (we noticed that a small portion of story plots generated by OpenPlot have missing outline bullet points, e.g., top-level point 1 is missing and the outline starts with second-level point 1a) to get 12824 plots as the training set, and performed SFT. In the training set, we use the premise as the prompt and the remaining parts of the story plots as the response. The end-to-end model is fine-tuned for 1000 steps with a mini-batch size of 8 using $8\times$ A100 80G GPUs simultaneously (so one step equals $8 \times 8 = 64$ story plots, and one epoch roughly equals 200 steps) from Llama2-7B-chat (Touvron et al., 2023) within a few hours.

To test the quality of story plots generated by our end-to-end model E2EPlot, we generate 500 premises, and for each premise, we generate a story plot pair, one from the DOC pipeline with Llama2 (i.e., OpenPlot) and the other from E2EPlot. We compare the overall quality of the 500 story plot pairs using GPT4 (OpenAI, 2023) as the reference. As Table 5 shows, the story plots generated by E2EPlot have comparable quality (slightly better) to the DOC pipeline with Llama2 (OpenPlot). Further details of evaluation by GPT4 are deferred to Appendix B.

| | DOC
(w/ OpenAI API) | OpenPlot
(DOC w/ LLama2) | E2EPlot
(SFT) |
|---|---|---|---|
| Generation speed | $\sim$ 5 mins
(with rate limit and several dollars) | $\sim$ 30 mins | $\sim$ 30 s |
| # Calls to LLM | $> 100$ | $> 100$ | 1 |

Table 1: Comparison of DOC, OpenPlot, and E2EPlot to generate one story plot.

Table 1 compares the performance of our end-to-end model E2EPlot and the previous DOC pipeline in generation procedure. In Section 3, we also provide an example of story plots generated from the same premise by OpenPlot and E2EPlot, respectively.

## 2.3 REWARD MODEL TRAINING AND RLHF: RLPLOT

One of the most important advantages of an end-to-end model such as our E2EPlot is that it can be easily fine-tuned from human feedback as long as a good reward model can be learned. With the pipeline in Section 2.1 using oasst-sft-6-llama-30b (Köpf et al., 2023) [3], we generate 7000 comparison story plot pairs, where each pair contains two story plots with the same premise. We use the same premise for each pair since it might be hard to compare two totally different story plots, and by controlling the premise, the preference labels might contain richer information related to our aspects of interest. Table 2 shows preference questions we ask human annotators to answer, where each question corresponds to one aspect of story quality. Note that there is no Q2 in Table 2 since Q2 asks annotators to explain their answer to Q1 with a minimum word count of 25, which is designed primarily to ensure the quality of the label.

**Remark 2.4.** *The following is an example of the answers to Q2: " In Plot A, there is a clear development of the relationship between the characters. It's obvious when Raven begins to learn about human emotion from James, and it's played out smoothly. In Plot B, Luna seems to have some repetitive plot points and the overarching plot is more about the human characters realizing that the android is intelligent. I prefer Plot A where Raven has the realization that she's massively impactful*

---

[3]We use oasst-llama-30b since when we produce the dataset containing 7000 comparison story plot pairs, Llama2 has not been released.

| | |
|---|---|
| **Q1** | Which story plot is *more interesting* to you overall? |
| **Q3** | In your opinion, which one of the plots above could generate a *more interesting book or movie* (when a full story is written based on it)? |
| **Q4** | Which story plot created more *suspense and surprise*? |
| **Q5** | Which story's characters or events do you *identify with or care for more*? |
| **Q6** | Which story has a *better ending*? |

Table 2: Five different questions for 7000 comparison story plot pairs.

*on the elderly man's life because of how close their relationship has become." The answer to Q2 is consistent with the answer to Q1 for the same annotator.*

Table 3 shows the human evaluation results for the 7000 story plot pairs. For each question (aspect), we keep the plot pairs that the response is either "Plot A is better" or "Plot B is better", and train a reward model from Llama2-7B-chat (Touvron et al., 2023) using cross-entropy loss for that specific aspect. Compared to Ammanabrolu et al. (2019), where the reward is based on how close the generated event is to a pre-trained goal, our reward model training is more similar to CARP (Castricato et al., 2022), a contrastively-trained preference model as a reward signal in story generation. Note that our reward model is trained on story plots instead of the whole story, which makes learning and human labeling easier without losing important signals. Finally, we obtain five different reward models along different aspects. Table 4 shows the validation accuracy of our trained reward models for each aspect, where the training-validation split ratio is $9 : 1$.

| Aspects | Plot A | Plot B | Both | Neither |
|---|---|---|---|---|
| **Q1** | 32% | 42% | 12% | 15% |
| **Q3** | 31% | 41% | 14% | 14% |
| **Q4** | 29% | 38% | 13% | 20% |
| **Q5** | 30% | 39% | 14% | 17% |
| **Q6** | 30% | 37% | 9% | 24% |

Table 3: Human preference labels on 7000 story plot pairs.

| Questions/aspects | **Q1** | **Q3** | **Q4** | **Q5** | **Q6** |
|---|---|---|---|---|---|
| Validation accuracy | 0.6050 | 0.5903 | 0.6365 | 0.5945 | 0.6723 |

Table 4: Validation accuracy for five different reward models on corresponding questions/aspects.

Using the trained reward models, we can further do RLHF on our end-to-end model `E2EPlot` to improve the quality of the generated story plots in various aspects. We use standard RLHF training objective (e.g., Equation (2) in Ouyang et al. (2022)):

$$\mathcal{L}_\phi = \mathbb{E}_{(x,y) \sim D_{\pi_\phi^{RL}}}[r_\theta(x, y) - \beta \log(\pi_\phi^{RL}(y|x)/\pi^{\text{SFT}}(y|x))]$$

where $\pi_\phi^{\text{RL}}$ is the learned policy by RLHF, $r_\theta$ is the reward model, $\pi^{\text{SFT}}$ is the policy after SFT (our `E2EPlot`), and $\beta$ is to control the KL divergence. The prompt $x$, i.e., the premise, is generated by the `OpenPlot` pipeline. We show the performance of our end-to-end model after RLHF (i.e., `RLPlot`) in Section 3.

## 3 RESULTS

In this section, we provide quantitative results on the quality of the generated story plots of our end-to-end models after SFT (`E2EPlot`) and RLHF (`RLPlot`). We also provide an example of the

story plot generated by `E2EPlot` for better visualization. Table 7 presents two story plots, where the left one is generated by `E2EPlot`, and the right one is generated by `OpenPlot` using the same premise. The two story plots are of similar quality, while our end-to-end model `E2EPlot` is much more efficient as shown in Section 2.2.

We show the quantitative results for `E2EPlot` in Section 3.1 and `RLPlot` in Section 3.2. The details of GPT4 evaluation is deferred to Appendix B.

## 3.1 PERFORMANCE OF END-TO-END MODEL AFTER SFT

| `OpenPlot` Wins | `E2EPlot` Wins | Ties |
|:---:|:---:|:---:|
| 45.8% | 46.8% | 7.4% |

Table 5: Comparison of generated story plots by `OpenPlot` and `E2EPlot` using GPT4 evaluation. We compare 500 story plot pairs where each pair has the same premise. The result shows that our end-to-end model `E2EPlot` can generate story plots of comparable quality to the DOC pipeline with Llama2.

Table 5 shows that our end-to-end model `E2EPlot` can generate story plots of comparable quality to the DOC pipeline with Llama2 (`OpenPlot`), at a much faster generation speed. Moreover, our end-to-end model can be easily fined-tuned from human feedback as shown in Section 3.2.

## 3.2 PERFORMANCE OF END-TO-END MODEL AFTER RLHF

| Aspects | `E2EPlot` Wins | `RLPlot` Wins | Ties |
|:---:|:---:|:---:|:---:|
| **Q1** | 44.0% | **54.0%** | 2.0% |
| **Q3** | **50.3%** | 46.0% | 3.7% |
| **Q4** | 39.3% | **60.0%** | 0.7% |
| **Q5** | 48.0% | **50.0%** | 2.0% |
| **Q6** | 42.3% | **53.7%** | 4.0% |

Table 6: Comparison of generated story plots by end-to-end SFT model (`E2EPlot`) and five different RLHF models (`RLPlot`) using GPT4 evaluation. For each row, we compare 300 story plot pairs where each pair has the same premise. We bold the results with a higher winning rate.

After SFT, we perform RLHF on our end-to-end model `E2EPlot` using five reward models for different aspects as discussed in Section 2.3 respectively, and obtain five models after RLHF. Models for Q1 to Q5 are trained for 20 steps, while the model for Q6 is trained for 40 steps, where each step corresponds to 64 story plots. The quality of models after RLHF (`RLPlot`) are shown in Table 6. For `RLPlot` for each aspect, we compare its quality to `E2EPlot` using 300 premises. For each premise, we let GPT4 compare which model generates a better story plot measured by the corresponding aspect.

Note that for some aspects (e.g., Q4, Q1, and Q6), the quality of the end-to-end model has improved after RLHF, while for other aspects (e.g., Q3), the model gets slightly worse. This might be because some aspects are more difficult to learn than others, which is also consistent with the result in Table 4 where Q3 has the lowest validation accuracy. It remains an important future direction to learn a more accurate reward model in a better way.

## 4 RELATED WORKS

**Automatic story generation.** Many previous works studying automatic story generation mainly focus on short stories with lengths of several sentences (Wang & Wan, 2019; Yao et al., 2019; Qin et al., 2019; Wang et al., 2022). Several recent works aim to generate long and coherent stories spanning thousands of words (Yang et al., 2022b;a; Zhou et al., 2023). All these works aim to

**Premise:** *"A teenage girl discovers a magical app that allows her to alter reality with just a few taps on her smartphone screen. However, she soon realizes that the app's magic comes at a terrible price."*

**Settings:** The story is set in the present day, in a small town in the United States.

**Characters:**

Ava Rose: Ava Rose is 17 years old and a tech-savvy teenager who loves nothing more than spending her free time exploring the latest apps and gadgets.
Elianore Starr: Elianore Starr is 28 years old and a brilliant app developer who has been working with Ava's older brother, Ethan, on a top-secret project.
...

**Outline:**
1. Ava discovers the magical app and begins to use it to alter reality, but she soon realizes that the app's magic comes at a terrible price. Scene: Characters: Ava Rose
    a. Ava discovers the app and starts to use it to improve her life and the lives of her friends. Scene: the town where Ava lives. Characters: Ava Rose
    b. Ava's friends become suspicious of her sudden changes and start to distance themselves from her. Scene: the town where Ava lives. Characters: Ava Rose
    ...
2. Ava confides in her best friend, Tess, about the app's dark side, and the two girls try to figure out a way to stop the app's power from consuming Ava's life. Scene: Characters: Ava Rose, Tess Sawyer
    ...
3. Elianore, the brilliant app developer, is recruited by Ava to help her fix the app and reverse its negative effects. Scene: Characters: Ava Rose, Elianore Starr
    ...
4. Ava and Elianore succeed in creating a new app that is free from the dark magic, but not before Ava's family and friends have suffered serious consequences as a result of their use of the app. Scene: Characters: Ava Rose, Elianore Starr
    ...
    d. Ava and her loved ones are finally safe from the app's power, but they will never be the same again. Scene: the town where Ava lives. Characters: Ava Rose

**Settings:** The story is set in the present day, in a small suburban town with a typical American high school.

**Characters:**

Jesse James: Jesse James is 17 years old, a high school student with a rebellious streak and a passion for photography.
Dean Defoe: Dean Defoe is 35 years old, a wealthy tech entrepreneur with a mysterious past.
...

**Outline:**
1. Jesse discovers the magical app and starts experimenting with it, creating small alterations to her reality without fully understanding the consequences. Scene: Characters: Jesse James
    a. Jesse discovers the app and starts using it to enhance her reality, but she soon realizes that the alterations are not temporary and begin to have unintended consequences. Scene: Jesse's bedroom. Characters: Jesse James
    b. Jesse's reality starts to change in unexpected ways, causing her to question her own perceptions and reality. Scene: Jesse's high school. Characters: Jesse James
    ...
2. As Jesse continues to use the app, the alterations become more profound and dangerous, revealing a darker truth about the app's magic and the people behind it. Scene: Characters: Jesse James
    ...
3. Jesse and her friends must work together to uncover the truth about the app, the entrepreneurs behind it, and the terrible price they are forcing upon the world. Scene: Characters: Jesse James
    ...
4. Jesse and her friends' investigations led them to uncover a shocking secret about the app that threatens to destroy the fabric of reality and undo all that they have ever known. Scene: Characters: Jesse James
    ...
    d. Jesse and her friends' investigations lead them to a shocking revelation that threatens to undo all they have ever known. Scene: a high-tech laboratory. Characters: Jesse James

Table 7: Story plots generated by `E2EPlot` and `OpenPlot` with the same premise. The left one is generated by our end-to-end model `E2EPlot`, and the right one is generated by the DOC pipeline with Llama2 (`OpenPlot`). Some contents of the plots are omitted for better visualization. One can see that the above two story plots are of similar quality.

generate a complete story, while our work focuses on story plot generation, which we conjecture is the most challenging and important part of story generation. In fact, lots of other works also recognize the importance of planning and thus either use story plots explicitly (Li et al., 2013; Fan et al., 2018; Yao et al., 2019; Goldfarb-Tarrant et al., 2020; Rashkin et al., 2020; Tian & Peng, 2022; Yang et al., 2022b;a) or their counterparts implicitly (Miao & Blunsom, 2016; Wang & Wan, 2019; Wang et al., 2022; Fan et al., 2019; Peng et al., 2018; Ippolito et al., 2019; Xu et al., 2020; Lin & Riedl, 2021) to improve generation quality. Regarding length and coherence, Yang et al. (2022a) is the most comparable to ours, although we use a more efficient end-to-end generator.

Beyond text-based story generation, some previous work also study relevant topics in the scope of multimodality, such as visual storytelling (sequential vision-to-language) (Ting-Hao et al., 2016), story visualization (paragraph to sequence of images) (Li et al., 2019) and StoryBench (text prompts to video stories) (Bugliarello et al., 2023).

**Prompt engineering.** To generate high-quality story plot datasets for training the end-to-end model as well as the reward models, we need a hard-wired pipeline to generate story plots by repeatedly prompting LLMs. Many previous works demonstrate the importance of prompt design (Brown et al., 2020; Zhong et al., 2021; Sanh et al., 2021; Lee et al., 2021; Ouyang et al., 2022; Wu et al., 2022; Kojima et al., 2022; Liu et al., 2023), and even for long-form generation, one important paradigm is to treat prompting as a subroutine and recursively prompt (Yang et al., 2022b;a; Zhou et al., 2023). Our pipeline for generating training datasets follows Yang et al. (2022a) while replacing OpenAI API calls with the Llama2 model, which eliminates the rate limit and makes large batch generation feasible.

**Long context generation.** Various recent efforts have focused on long context generation either by extending the length of the context window (Haviv et al., 2022; Sun et al., 2022b; Press et al., 2022; Chen et al., 2023a) or improving efficiency for long context generation (Child et al., 2019; Kitaev et al., 2020; Choromanski et al., 2022; Zhang et al., 2023). Although the length of the outputs has been extended, these mechanisms do not necessarily guarantee the quality of the generated context, especially for story generation. Besides mechanical modifications, some works achieve long context generation in a hierarchical manner (Fan et al., 2018; Yao et al., 2019; Fan et al., 2019; Tan et al., 2021; Sun et al., 2022a; Yang et al., 2022a; Zhou et al., 2023). In this paper, we avoid generating super long outputs by focusing on generating plots, which typically consist of around 1k tokens. However, after the plot generation stage, the above mechanisms for extending context length or accelerating inference would be helpful in generating a complete long story given the plot.

**Human-in-the-loop story generation.** Different from automatic story generation, several previous works use human-in-the-loop methods to generate long stories (Goldfarb-Tarrant et al., 2019; Coenen et al., 2021; Lee et al., 2022; Chung et al., 2022; Ippolito et al., 2022; Mirowski et al., 2023). Note that although our end-to-end generator is totally automatic without human intervention, it is easy for humans to co-create after the plot generation stage. Since the story plot is relatively short and thus easy for humans to evaluate, one can easily edit the plot to obtain a more desired story.

## 5 CONCLUSIONS

In this work, we study end-to-end story plot generation. By improving the previous story plot generation pipeline and obtaining high-quality training data, we successfully train an end-to-end model, which is able to generate story plots of comparable quality to the most advanced methods to date much more efficiently. Moreover, due to the end-to-end nature of our method, we further fine-tune it with human feedback, which improves the model for different aspects.

There are many important and interesting future directions. For example, the current generation speed is around 30 seconds, which is still somewhat slow from a user experience perspective. It might be possible to incorporate techniques for more efficient inference such as Zhang et al. (2023) into our end-to-end model. Also, since the previous DOC pipeline has more flexibility for controlling the level of granularity, it would be appealing to train an end-to-end model that inherits this property. Additionally, a high-quality reward model is important to improve the end-to-end generator.

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

# A    DETAILED PROMPTS IN SECTION 2.1.2

In this section, we provide the details of the whole rebuilt DOC pipeline, `OpenPlot`, including the exact format of the main prompts. Note that some comments and remarks have already appeared in Section 2.1.2. We keep those repeated contents in this section so that readers can find all important and necessary information in this section without referring back to Section 2.1.2.

**Premise.**    The first step is to generate a premise for the story. We use the following prompt:

> Write a premise for a short story in one paragraph with two to three sentences.
>
> Premise:

Note that we enforce that the language model's response starts after "Premise:". This structure precludes non-informative responses such as "Sure!", "No problem!" and ensures that the format of the output is structured and thus easy to process.

**Setting.**    After obtaining the premise of the whole story, we also infer the setting of the story from the premise using the following prompt:

> Premise: PREMISE
>
> Describe the setting of the story.
>
> The story is set in

Here, PREMISE is the premise of the story, which is the output of the previous prompt. For example, PREMISE could be *"Alex dreams of becoming a famous writer but becomes trapped in a time warp, reliving the same day over and over."* The premise and the setting build the most basic skeleton of the story. In the following parts, we use SETTING to denote the content of the settings, such as *"The story is set in a world where magic is real and has been suppressed by a powerful and corrupt government."*

**Characters.**    One of the most important elements in a story is the characters, whose intrinsic motivation and interactions with each other are vital to the trajectories of the whole story. If one views the whole story as an RL environment, the characters in the story play essentially the same role as agents in RL. Based on the premise and settings, we use the following prompts to generate characters one by one:

> Premise: PREMISE
>
> Setting: SETTING
>
> List the names and details of all major characters.
>
> 1.
>
> Full Name:

> Premise: PREMISE
>
> Setting: SETTING
>
> List the names and details of all major characters.
>
> 1.
>
> Full Name: NAME_1
>
> Use ONLY one short sentence for the following description relevant to the story, focusing on relationship between characters, occupation and experience instead of appearance. Only ONE sentence is allowed!

> Character Portrait: NAME_1 is

The first prompt helps to generate the name of the next character. We use NAME_$k$ to denote the full name of the $k$-th character (e.g., NAME_1 can be "Jack Thomas"). After obtaining the name of the $k$-th character, we use the second prompt to generate the character portrait for that character.

**Remark A.1.** *Note that in the second prompt, we added detailed instructions on the generated portrait. In the original DOC code (Yang et al., 2022a), this instruction is unnecessary. However, when we replace the text-davinci-002 engine with the open-sourced Llama2-13B-chat model, due to the performance difference between these two models, the Llama2 model tends to output a longer description for the portrait without any instruction. Moreover, without instruction, the output of Llama2 focuses more on the age appearance of the character instead of occupation, experiences, or relationship with other characters (e.g., "Tom is 22 years old and has brown curly hair"). After adding the above instruction, the output of the second prompt tends to contain more compact and dense information about the character.*

After generating the first character, we can repeat the above procedure to generate more characters. For example, the following prompt continues to generate the full name of the second character:

> Premise: PREMISE
>
> Setting: SETTING
>
> List the names and details of all major characters.
>
> 1.
> Full Name: NAME_1
>
> Character Portrait: PORTRAIT_1
>
> 2.
> Full Name:

We can control the number of major characters in the story, and in our implementation, we set the desired number to be 3-6, since too few characters are likely to make the story boring, and too many characters will reduce opportunities for characters to interact with each other.

**Outline.** After generating all the major characters, we are ready to build the main skeleton of the story. Same as Yang et al. (2022a), we aim to create a hierarchical outline. In this paper, we create two-level hierarchical outlines, where the top level typically contains four bullet points, and the second level contains three to four subpoints under each top-level point. We also generate the outline in breadth-first order, i.e., we first generate the top-level outline (numbered as 1, 2, 3, . . .) and then generate the sub-level outline (numbered as 1a, 1b, . . ., 2a, . . .). The following prompt can be used to generate the first point of the top-level outline, where CHARACTERS contains the name and portrait of all major characters.

> Premise: PREMISE
>
> Setting: SETTING
>
> Characters: CHARACTERS
>
> Outline the main plot points of the story using no more than 4 points, generating one point at a time. IMPORTANT: Please make sure that the story has a clear end at or before Point 4.
>
> 1.

**Remark A.2.** *To make the generated story plots more reasonable, we added corresponding instructions. First, to control the length of the top-level outline, we require the LLM to use no more than 4 points. Second, to make the generation procedure more stable, we require the LLM to generate only one point at a time. Third, during the preliminary experiments, we found that the generated plots*

*sometimes have a missing ending. To address this issue, we explicitly ask the LLM to make sure that the generated top-level outline has a clear ending. A tricky part is that adding "IMPORTANT: Please" significantly improves the quality of the generated outline in terms of a clear ending.*

After generating the top bullet point 1, we can include the content of point 1 in the prompt and continue to generate the subsequent points until the whole top-level outline is complete. Since we generate the outline in breadth-first order, we will expand each top-level node in sequence. Note that it is important to keep the generated subpoint consistent with not only the previous points but also the subsequent points. To achieve this, Yang et al. (2022a) use a completion model (text-davinci-002) and add a suffix containing the content of all subsequent points. For example, to generate subpoint 1a, they use the prompt of the following format:

> Premise: PREMISE
>
> Setting: SETTING
>
> Characters: CHARACTERS
>
> Outline:
>
> 1. POINT_1
>
> List the main events that occur under this heading using no more than 4 points, starting from the beginning, generating one or two points without repeating the content of the suffix and stop.
>
>     a.
>
> SUFFIX:
>
>     b. POINT_2
>
>     c. POINT_3
>
>     d. POINT_4

The actual prompt in the above box is the content before "SUFFIX", and the content after "SUFFIX" can be passed to the completion model as an argument. We use POINT_$k$ to denote the content of the top-level bullet point $k$, which can be, e.g., "Luke discovers the mysterious box in his grandfather's attic and initially dismisses it as a strange trinket with no value."

**Remark A.3.** *There is a trick to make the generated content more consistent with the existing content by shifting the points after the current point to a lower level, e.g., top-level point 2 is shifted to sub-level point 1b. Also, detailed instruction on generating the current sub-level point is not necessary for a completion model such as text-davinci-002, but is helpful when we call a chat model such as Llama2. We will discuss this in detail in Remark A.4.*

Since we use Llama2 in our pipeline, which is a chat model and thus does not accept suffixes, we need to simulate a completion model using a chat model. For convenience, we let CONTENT_OF_PREFIX and CONTENT_OF_SUFFIX denote the content before and after "SUFFIX" in the above box respectively. Then one can use the following prompt to simulate a completion model:

> Imagine you are a text completion robot. Give the output of the following task with the given suffix and prompt. Please follow the instructions below.
>
> Instructions: Your output should not contain the content of the suffix. Only use the suffix as complementary information. The output should mainly be based on the prompt. Now the suffix begins.
>
> Suffix:
>
> CONTENT_OF_SUFFIX
>
> End of Suffix

> Now the prompt begins.
>
> Prompt:
>
> CONTENT_OF_PREFIX

**Remark A.4.** *First, note that we put the content of the prefix after the suffix to make it easier for the LLM to continue with the prefix. We also need to add instructions to obtain desired results since otherwise, the LLM will memorize the context of the suffix, and the output will be nearly identical to the suffix. In Remark A.3, we also mentioned that there is a detailed instruction in the prefix on how to generate the current sub-level points, which can "reinforce" the requirement that the output should not repeat the content of the suffix.*

After expanding each top-level point by generating sub-points, we obtain a hierarchical outline, and the story plot generation procedure is finished.

# B   ADDITIONAL DETAILS OF GPT4 EVALUATION

We use the following prompt (adapted from Zheng et al. (2023)) to compare two story plots with the same premise:

> Please act as an impartial judge and evaluate the quality of the story plots generated by two AI models. The two story plots have the same premise. You should choose the story plots that have better qualities. **Your evaluation should focus on the overall qualities.** Begin your evaluation by comparing the two story plots and provide a short explanation. Avoid any position biases and ensure that the order in which the story plots were presented does not influence your decision. Do not allow the length of the story plots to influence your evaluation. Be as objective as possible. After providing your explanation, output your final verdict by strictly following this format: "[[A]]" if story plot A is better, "[[B]]" if story plot B is better, and "[[C]]" for a tie.
>
> [The Start of story plot A]
>
> STORY_PLOT_A
>
> [The End of story plot A]
>
> [The Start of story plot B]
>
> STORY_PLOT_B
>
> [The End of story plot B]

We use the above prompt to compare the overall qualities of `E2EPlot` and `OpenPlot`. When comparing `E2EPlot` and `RLPlot`, we replace the sentence "Your evaluation should focus on the overall qualities" with the corresponding aspect, e.g., "Your evaluation should focus on the Aspect: Which story plot created more suspense and surprise?".

For each comparison, we randomly shuffle the position of the two plots to avoid position bias. Below, we provide an example of the response of GPT4:

> After a careful comparison of both story plots, it is clear that story plot B generates more suspense and surprise elements.
>
> In story plot A, there are a few suspenseful elements like the discovery of the hidden room in the attic and basement, and the revelation of the murderer's identity. However, these incidents are predictable to an extent as they are common tropes in detective stories. Moreover, the identity of the murderer, a member of the Blackwood family, is revealed without a significant build-up which reduces the element of surprise.
>
> Contrarily, in story plot B, there are several suspenseful and surprising twists. Initially, Gertrude Rutledge, the mansion's former housekeeper, becomes the prime suspect. The

story then reveals her as a ghost causing the disturbances. As the plot unfolds, there's a surprising revelation about Malcolm Crawford's past and his connection to the haunting. There is also the unexpected element of Gertrude Rutledge having an accomplice in the murder and haunting. Each of these plot points introduces unexpected turns and secrets that enhance the suspense and surprise of the story.

Therefore, the final verdict favors story plot B for its superior ability to create more suspense and surprise. [[B]]

## C  HUMAN EVALUATION

In this section, we provide human evaluation results comparing the quality of `OpenPlot`, `E2EPlot` and `RLPlot`. Since we have five `RLPlot` models, we name them `RLPlot_Q1`, `RLPlot_Q3`, `RLPlot_Q4`, `RLPlot_Q5`, and `RLPlot_Q6`, respectively. We sent 500 (`OpenPlot` vs. `E2EPlot`) + 300 * 5 (`RLPlot` vs. `E2EPlot`) = 2000 story plot pairs, which are the same as what we used for GPT4 evaluation, to the prolific platform, and around 1700 of them are labeled (each pair requires one label, each participant are required to label five pairs, and some of the participants labeled part of the five pairs). For each pair, the participants are required to answer seven questions, where Q1-Q6 are the same as in Table 2, Q2 is a free text explanation for Q1, and Q7 is 'Which story is better in overall quality'. The result is shown in Appendix C.1.

### C.1  HUMAN EVALUATION RESULTS

The result for human evaluation is presented in Tables 8 to 13.

|    | OpenPlot wins | E2EPlot wins | Tie |
|----|----------------|----------------|------|
| **Q1** | 41.2% | 43.4% | 15.4% |
| **Q3** | 40.5% | 40.5% | 19.0% |
| **Q4** | 43.2% | 40.7% | 16.1% |
| **Q5** | 40.2% | 39.5% | 20.2% |
| **Q6** | 39.0% | 39.8% | 21.2% |
| **Q7** | 38.5% | 41.2% | 20.2% |

Table 8: Comparison of `OpenPlot` and `E2EPlot` on 410 story plot pairs (500 sent and 410 labeled) by human evaluation

|    | RLPlot_Q1 wins | E2EPlot wins | Tie |
|----|----------------|----------------|------|
| **Q1** | 48.8% | 37.9% | 13.3% |
| **Q3** | 44.8% | 38.3% | 16.9% |
| **Q4** | 46.8% | 32.3% | 21.0% |
| **Q5** | 40.7% | 35.1% | 24.2% |
| **Q6** | 47.6% | 35.1% | 17.3% |
| **Q7** | 43.1% | 36.3% | 20.6% |

Table 9: Comparison of `RLPlot_Q1` and `E2EPlot` on 248 story plot pairs (300 sent and 248 labeled) by human evaluation

### C.2  COMPLEMENTARY GPT4 EVALUATION RESULT

For completeness, we also provide corresponding evaluation results by GPT4. The result for GPT4 evaluation is presented in Tables 14 to 19.

|     | RLPlot_Q3 wins | E2EPlot wins | Tie   |
| --- | -------------- | ------------ | ----- |
| **Q1** | 41.5%          | 41.1%        | 17.4% |
| **Q3** | 38.2%          | 41.1%        | 20.7% |
| **Q4** | 34.4%          | 45.6%        | 19.9% |
| **Q5** | 35.3%          | 38.6%        | 26.1% |
| **Q6** | 39.4%          | 38.2%        | 22.4% |
| **Q7** | 36.1%          | 41.9%        | 22.0% |

Table 10: Comparison of `RLPlot_Q3` and `E2EPlot` on 241 story plot pairs (300 sent and 241 labeled) by human evaluation

|     | RLPlot_Q4 wins | E2EPlot wins | Tie   |
| --- | -------------- | ------------ | ----- |
| **Q1** | 41.3%          | 40.9%        | 17.9% |
| **Q3** | 42.5%          | 39.3%        | 18.3% |
| **Q4** | 40.1%          | 38.5%        | 21.4% |
| **Q5** | 37.7%          | 38.5%        | 23.8% |
| **Q6** | 40.1%          | 38.9%        | 21.0% |
| **Q7** | 40.1%          | 38.1%        | 21.8% |

Table 11: Comparison of `RLPlot_Q4` and `E2EPlot` on 252 story plot pairs (300 sent and 252 labeled) by human evaluation

|     | RLPlot_Q5 wins | E2EPlot wins | Tie   |
| --- | -------------- | ------------ | ----- |
| **Q1** | 43.4%          | 43.0%        | 13.5% |
| **Q3** | 45.0%          | 42.6%        | 12.4% |
| **Q4** | 46.2%          | 42.6%        | 11.2% |
| **Q5** | 43.8%          | 37.5%        | 18.7% |
| **Q6** | 41.8%          | 38.2%        | 19.9% |
| **Q7** | 43.0%          | 37.8%        | 19.1% |

Table 12: Comparison of `RLPlot_Q5` and `E2EPlot` on 251 story plot pairs (300 sent and 251 labeled) by human evaluation

|     | RLPlot_Q6 wins | E2EPlot wins | Tie   |
| --- | -------------- | ------------ | ----- |
| **Q1** | 38.7%          | 44.0%        | 17.3% |
| **Q3** | 35.9%          | 44.8%        | 19.4% |
| **Q4** | 39.5%          | 41.1%        | 19.4% |
| **Q5** | 35.1%          | 39.1%        | 25.8% |
| **Q6** | 36.3%          | 45.2%        | 18.5% |
| **Q7** | 35.9%          | 41.1%        | 23.0% |

Table 13: Comparison of `RLPlot_Q6` and `E2EPlot` on 248 story plot pairs (300 sent and 248 labeled) by human evaluation

|    | OpenPlot wins | E2EPlot wins | Tie |
|----|------|------|------|
| **Q1** | 53.2% | 45.2% | 1.6% |
| **Q3** | 53.6% | 40.6% | 5.8% |
| **Q4** | 53.2% | 45.6% | 1.2% |
| **Q5** | 53.0% | 45.2% | 1.8% |
| **Q6** | 48.2% | 50.4% | 1.4% |
| **Q7** | 45.8% | 46.8% | 7.4% |

Table 14: Comparison of OpenPlot and E2EPlot on 500 story plot pairs by GPT4 evaluation

|    | RLPlot_Q1 wins | E2EPlot wins | Tie |
|----|------|------|------|
| **Q1** | 54.0% | 44.0% | 1.6% |
| **Q3** | 53.0% | 41.0% | 6.0% |
| **Q4** | 49.0% | 50.7% | 0.3% |
| **Q5** | 51.7% | 46.7% | 1.7% |
| **Q6** | 56.7% | 41.0% | 2.3% |
| **Q7** | 49.7% | 44.3% | 6.0% |

Table 15: Comparison of RLPlot_Q1 and E2EPlot on 300 story plot pairs by GPT4 evaluation

|    | RLPlot_Q3 wins | E2EPlot wins | Tie |
|----|------|------|------|
| **Q1** | 45.3% | 52.0% | 2.7% |
| **Q3** | 46.0% | 50.3% | 3.7% |
| **Q4** | 43.3% | 56.3% | 0.3% |
| **Q5** | 44.0% | 55.0% | 1.0% |
| **Q6** | 49.7% | 47.3% | 3.0% |
| **Q7** | 46.7% | 44.3% | 9.0% |

Table 16: Comparison of RLPlot_Q3 and E2EPlot on 300 story plot pairs by GPT4 evaluation

|    | RLPlot_Q4 wins | E2EPlot wins | Tie |
|----|------|------|------|
| **Q1** | 55.0% | 43.0% | 2.0% |
| **Q3** | 51.7% | 42.0% | 6.3% |
| **Q4** | 60.0% | 39.3% | 0.7% |
| **Q5** | 51.7% | 45.7% | 2.7% |
| **Q6** | 58.7% | 40.3% | 1.0% |
| **Q7** | 44.7% | 46.0% | 9.3% |

Table 17: Comparison of RLPlot_Q4 and E2EPlot on 300 story plot pairs by GPT4 evaluation

|    | RLPlot_Q5 wins | E2EPlot wins | Tie |
|----|------|------|------|
| **Q1** | 40.3% | 56.0% | 3.7% |
| **Q3** | 42.0% | 53.0% | 5.0% |
| **Q4** | 41.7% | 57.0% | 1.3% |
| **Q5** | 50.0% | 48.0% | 2.0% |
| **Q6** | 51.0% | 47.3% | 1.7% |
| **Q7** | 43.3% | 49.7% | 7.0% |

Table 18: Comparison of RLPlot_Q5 and E2EPlot on 300 story plot pairs by GPT4 evaluation

|    | `RLPlot_Q6` wins | `E2EPlot` wins | Tie  |
|----|------------------|----------------|------|
| **Q1** | 44.3% | 53.7% | 2.0% |
| **Q3** | 42.7% | 53.7% | 3.7% |
| **Q4** | 44.7% | 55.0% | 0.3% |
| **Q5** | 42.3% | 54.3% | 3.3% |
| **Q6** | 53.7% | 42.3% | 4.0% |
| **Q7** | 44.7% | 50.0% | 5.3% |

Table 19: Comparison of `RLPlot_Q6` and `E2EPlot` on 300 story plot pairs by GPT4 evaluation

