# OpenReview forum: "End-to-end Story Plot Generator"
_ICLR.cc/2024/Conference — Submitted to ICLR 2024_

### Official Review · Reviewer_G5uN · 2023-10-28

**Soundness:** 2 fair
**Presentation:** 4 excellent
**Contribution:** 3 good
**Rating:** 6
**Confidence:** 3

**Summary:**

This paper introduces an end-to-end model for the task of (text) story plot generation. The authors first replicate a previous model for story generation (DOC) with open-source architectures and show challenges and fixes to overcome some of the challenges in the transition. Then, they use this model to create a large number of story plots that are then used to fine-tune another, 7B model for this task, showing competitive performance (according to GPT-4) with the teacher model. Finally, the authors collect human preferences from plots generated with the same premise and further tune the fine-tuned model with RLHF, resulting in better performance across 5 metrics (again, according to GPT-4).

**Strengths:**

1. The task of story plot generation is interesting and allows to disentangle two major phases of content creation often seen in humans: planning and coarse-to-fine generation
2. The authors show the intricacies of adapting a previous model from closed-source APIs to open-source alternatives that do not have the same capabilities (completion/infilling vs. chat)
3. The authors can train a 7B model that is relatively fast and seems to generate stories of similar quality as a 13B model that follows the original protocol with 100s of calls.
4. The paper is well written and easy to follow

**Weaknesses:**

1. My major concern is the lack of human evaluation. The authors state multiple times that “story plot is relatively short and thus easy for humans to evaluate” yet perform no human evaluation. Relying solely on a model like GPT-4 for such a complex task is, in my opinion, a major limitation to the soundness of the claims. I hence invite the authors to pair the main evaluations (ie Table 5 and 6) with human evaluation. I will then happily advocate for acceptance.
2. Another minor limitation is that RLHF results in 5 different models, which are each compared against a single SFT model. Could the authors consider combining the different rewards into a single RLHF model and then ask humans to compare it against SFT according to the metrics Q1-6?

---
Rebuttal: The authors added human evaluation, and said that (2) is a direction for future work.

**Questions:**

1. The Related Work section is comprehensive. To make the paper’s scope broader and linked to work in the computer vision and multimodal communities, I would recommend adding a brief paragraph on work in story generation for image-to-text [1], text-to-image [2] and text-to-video [3] tasks.
2. Are the results in Table 4 a “validation” of the same RLHF training story plots?
3. In page 2, when you say “a completion model which accepts a suffix” I was not entirely sure what you meant. It then became clear later throughout the paper. I recommend adding a brief explanation that you mean a model that can do text infilling given a text prefix and a text suffix, and cite [4].
4. Line 3 of Sec 3 should reference Table 7, not Table 1.
5. First line of Sec 3.2 might make it explicit that the RLHF results in 5 models.

---

[1] Huang et al. Visual Storytellin. NAACL’16

[2] Li et al. StoryGAN: A sequential conditional gan for story visualization. CVPR’19

[3] Bugliarello et al. StoryBench: A Multifaceted Benchmark for Continuous Story Visualization. arXiv 2308.11606

[4] Bavarian et al. Efficient Training of Language Models to Fill in the Middle. arXiv 2207.14255

---

> ### Author Response · Authors · 2023-11-20
>
> We thank the reviewer for their helpful and insightful comments. Below we respond to the questions respectively.
>
> > My major concern is the lack of human evaluation… I hence invite the authors to pair the main evaluations (ie Table 5 and 6) with human evaluation. I will then happily advocate for acceptance.
>
> Thanks for your suggestion. We added human evaluation results, and the details can be found in the global response.
>
> >  Another minor limitation is that RLHF results in 5 different models, which are each compared against a single SFT model. Could the authors consider combining the different rewards into a single RLHF model and then ask humans to compare it against SFT according to the metrics Q1-6?
>
> Training a unified model that can produce better story plots in multiple aspects simultaneously would definitely be an interesting future direction, which is also studied in a concurrent work [1]. Note that according to our human evaluation results, the model RLPlot_Q1 achieves better performance than E2EPlot consistently in all the aspects, which shows that even if we only use a reward model for a specific aspect, it might generalize well to other aspects since usually different aspects might be correlated.
>
> > Q1. The Related Work section is comprehensive. To make the paper’s scope broader and linked to work in the computer vision and multimodal communities, I would recommend adding a brief paragraph on work in story generation for image-to-text [1], text-to-image [2] and text-to-video [3] tasks.
>
> Thanks for your suggestion. We have added a brief paragraph on these three works in the revision.
>
> > Q2. Are the results in Table 4 a “validation” of the same RLHF training story plots?
>
> The 7000 story pair preference dataset was split into training and validation sets with a ratio of 9:1 for each one of the five reward models. We have made it clear in the revision.
>
> > Q3 .In page 2, when you say “a completion model which accepts a suffix” I was not entirely sure what you meant. It then became clear later throughout the paper. I recommend adding a brief explanation that you mean a model that can do text infilling given a text prefix and a text suffix, and cite [4].
>
> Thanks for your suggestion. We have fixed it in the revision.
>
> > Q4. Line 3 of Sec 3 should reference Table 7, not Table 1.
>
> Thanks for catching this! We have corrected it in the revision.
>
> > Q5. First line of Sec 3.2 might make it explicit that the RLHF results in 5 models.
>
> Thanks for your suggestion. We explicitly mentioned that RLHF results in 5 models in the revision.
>
>
> **References:**
>
> [1] Learning Personalized Story Evaluation, https://aps.arxiv.org/abs/2310.03304.

---

> > ### Comment · Reviewer_G5uN · 2023-11-23
> >
> > Thank you for addressing my questions. I have increased my score.
> > I noticed several concerns raised by other reviewers, with which I'll engage during reviewer discussion.

---

### Official Review · Reviewer_vgZV · 2023-10-31

**Soundness:** 1 poor
**Presentation:** 2 fair
**Contribution:** 1 poor
**Rating:** 1
**Confidence:** 5

**Summary:**

This paper attempts to replicate the story outline generation by DOC (Yang et al. 2022a), which makes use of GPT-3. Instead, this paper replaces GPT-3 with the opensource Llama2-13B. In addition, it performs an end-to-end finetuning on Llama2-7B, achieving speedup over repeated API calls. Finally, it performs RLHF on a collected dataset of human feedback.

Given the low-quality author response, I decided to lower my score to strong reject.

**Strengths:**

The paper is generally well written, if missing a few technical definitions here and there.

The speed-up in Table 1 is significant.

The human comparison data of the 7000 story pairs would be quite interesting, if released.

**Weaknesses:**

For someone who is familiar with DOC and LLMs, the paper does not seem to offer any new insight or new knowledge. Yes, Llama-2 and GPT-3 can do roughly the same things. Yes, we can use a supervisedly finetuned model to replace repeated LLM calls. Yes, we can do RLHF. All these are common knowledge. Hence, it may seem that the paper does not make a real scientific contribution. It may be an interesting engineering effort, but may not qualify as a scientific publication.

The decision to use GPT-4 as the evalution for final story quality seems dubious. The paper has not offered any evidence that GPT-4 is good at the task. The paper makes the claim that the RLHF stories are better at suspense and surprise. However, do we know if GPT-4 is good at detecting suspense or surprise?

Since the author has spent significant effort to collect human ratings on 7000 story pairs, why not do another 300 pairs? This would create a much more solid evaluation.

How is story generation different from other forms of structured text generation, such as poetry or argumentative essays? The paper has used specific prompts to handle aspects of stories such as characters or settings. But it is not explicit that if there is any principle behind the writing of these prompts or if they solely rely on trial and error by the paper authors. This goes back to the question: what is it that we can learn from this paper?

Minor comments:

Section 2.1.1 The authors do not define what is meant by "supports suffix". I managed to guess the meaning from the context but it created temporary confusion.

The authors make several claims about how humans supposedly do things. Humans write stories in a specific way (Page 2, Paragarph 2). Humans write from coarse to fine (Challenge 1). However, these claims are unsubstantiated.

**Questions:**

- What novel or surprising scientific insights or findings are reported by this paper?
- What evidence can support the claim that humans write stories by first planning an outline?

---

> ### Author Response · Authors · 2023-11-20
>
> We thank the reviewer for their helpful and insightful comments. Below we respond to the questions respectively.
>
> > For someone who is familiar with DOC and LLMs, the paper does not seem to offer any new insight or new knowledge. Yes, Llama-2 and GPT-3 can do roughly the same things. Yes, we can use a supervisedly finetuned model to replace repeated LLM calls. Yes, we can do RLHF. All these are common knowledge. Hence, it may seem that the paper does not make a real scientific contribution.
>
> As reviewer zk9y mentioned, Our paper “addresses a difficult problem, which has a long history in AI and to which the advent of LLM offers new perspectives”, Also, our work “contains a number of original aspects (experimental approach, evaluation)” and “includes a transparent account of prompt engineering aspects faced during the development of the work, which might be beneficial to readers”. We discussed this and our contribution in the **Novelty** section in the global response in detail. Also, studying the e2e **plot** generator instead of the whole story generator provides important insight into the story generation problem. See the **Why story plot rather than the entire story** section in the global response for more details.
>
> > Since the author has spent significant effort to collect human ratings on 7000 story pairs, why not do another 300 pairs? This would create a much more solid evaluation.
>
> Thanks for your suggestion. We added human evaluation results during the rebuttal to make the evaluation more solid. Please see the global response for details.
>
> > How is story generation different from other forms of structured text generation, such as poetry or argumentative essays? The paper has used specific prompts to handle aspects of stories such as characters or settings. But it is not explicit that if there is any principle behind the writing of these prompts or if they solely rely on trial and error by the paper authors.
>
> The reason that we use the (premise, settings, characters, outlines) structure of the plot is as follows (also discussed in the response to reviewer zk9y).
>
> - The underlying principle is “coarse-to-fine”, i.e., to progressively provide more and more detailed information. This coarse-to-fine representation makes end2end autoregressive generation easier.
> - While short, this representation still contains a good amount of information about the story, which is helpful for multiple perspective evaluations (represented by multiple questions).
> - This representation is used by the SOTA story generation work DOC [1] for the planning stage. It can be used to generate high-quality full-length stories. This demonstrates that such a structure can lead to full story generation.
> - The plot structure we chose in this paper is text-based, which could greatly benefit from the powerful text-based LLMs that take the plot as input and produce the whole story (usually with multiple steps such as the second stage of DOC [1]).
>
> Other forms of structured text generation, like poetry or argumentative essays, may not require that many levels of hierarchy.
>
> We acknowledge that we haven’t further leveraged the special structure of the story yet (e.g., characters) in our generation process, which will be our future work.
>
> > Section 2.1.1 The authors do not define what is meant by "supports suffix". I managed to guess the meaning from the context but it created temporary confusion.
>
> We have fixed it in the revision to avoid confusion. Thanks for your comment.
>
>
> > The authors make several claims about how humans supposedly do things. Humans write stories in a specific way (Page 2, Paragarph 2). Humans write from coarse to fine (Challenge 1). However, these claims are unsubstantiated.
>
> The planning-based method is common in many generation tasks (e.g., [3],[4]).  The coarse-to-fine generation has been widely used in previous work (such as event-to-sentence generation in [2], which is mentioned by reviewer zk9y).  Also, it is supported by the comment of reviewer G5uN that “The task of story plot generation is interesting and allows to disentangle two major phases of content creation often seen in humans: planning and coarse-to-fine generation”.
>
>
>
> > What novel or surprising scientific insights or findings are reported by this paper?
>
> We discussed this in detail in the global response (see the **Novelty** section in the global response).

---

> > ### Author Response · Authors · 2023-11-20
> >
> > **References:**
> >
> > [1] Yang, Kevin, Dan Klein, Nanyun Peng, and Yuandong Tian. "Doc: Improving long story coherence with detailed outline control." arXiv preprint arXiv:2212.10077 (2022).
> >
> > [2] Ammanabrolu, Prithviraj, Ethan Tien, Wesley Cheung, Zhaochen Luo, William Ma, Lara Martin, and Mark Riedl. "Guided neural language generation for automated storytelling." In Proceedings of the Second Workshop on Storytelling, pp. 46-55. 2019.
> >
> > [3] Tian, Yufei, and Nanyun Peng. "Zero-shot sonnet generation with discourse-level planning and aesthetics features." arXiv preprint arXiv:2205.01821 (2022).
> >
> > [4] Zhang, Zhexin, Jiaxin Wen, Jian Guan, and Minlie Huang. "Persona-Guided Planning for Controlling the Protagonist's Persona in Story Generation." arXiv preprint arXiv:2204.10703 (2022).

---

> > > ### Comment · Reviewer_vgZV · 2023-11-22
> > >
> > > I remain unconvinced. I fully agree that story generation is a difficult problem. However, the paper does not address any of the difficult aspects of story generation. Instead, it achieves some speed-up. Who ever complained about their stories not being generated fast enough?
> > >
> > > > The planning-based method is common in many generation tasks (e.g., [3],[4]). The coarse-to-fine generation has been widely used in previous work (such as event-to-sentence generation in [2], which is mentioned by reviewer zk9y). Also, it is supported by the comment of reviewer G5uN that “The task of story plot generation is interesting and allows to disentangle two major phases of content creation often seen in humans: planning and coarse-to-fine generation”.
> > >
> > > This is entirely missing the point. Computers doing something in no way implies that humans do this too. A comment of a reviewer cannot be taken as scientific evidence.

---

> > > > ### Author Response · Authors · 2023-11-22
> > > >
> > > > We thank the reviewer for the time spent reading our response. Below we provide further response.
> > > >
> > > > **[Focusing on story plot as our Novelty]**  We are happy to see that the reviewer agrees that story generation is a difficult problem. We respectfully disagree that ``the paper does not address any of the difficult aspects of story generation''. Instead, achieving an end-to-end style of (story plot) generation is a very important step from a hand-crafted story generation method to a highly automatic method. One of the important contributions of our work is that we propose focusing on the *story plot*, which carries the most essential information of the whole story while being much shorter, instead of focusing on generating a whole story using an end-to-end style that is over-ambitious.
> > > >
> > > > Knowing that ``we can use a supervisedly finetuned model to replace repeated LLM calls'' is easy, but the challenging part is **what** SFT (or e2e) model  (the whole story or the story plot or others) should we aim to learn. In our paper, we answered this question and implemented it by successfully fine-tuning an end-to-end story plot generator.
> > > >
> > > > **[The importance of a fast and end-to-end story plot generator]** Here are the advantages of an end-to-end model:
> > > > 1. A significant speed-up.
> > > > 2. It can be easily improved by human-labeled data, which is very hard or even impossible for previous hand-crafted non-differentiable methods.
> > > >
> > > > Note that we disagree with the reviewer that “Who ever complained about their stories not being generated fast enough?”. While human writers may not complain about it, for AI to write a good story, a key ingredient is to (1) generate lots of candidate stories, (2) evaluate them quickly with (semi)-automatic metrics and (3) improve the models based on the feedback. For (1) and (2), we really need a fast generator for story plots, e.g., for (2) when a user plays with an automatic story generation model, it is embarrassing to let the user wait for five minutes to see even a single story plot.
> > > >
> > > > **[Justification of planning-based/coarse-to-fine methods]** For the planning-based method or coarse-to-fine method, we agree that not everyone will adopt such a strategy. However, this is a common strategy that many authors may use, including an amateur novel writer in our team. More importantly, these methods are proven to be effective in the generative method in the story generation literature (e.g., [3][4], also referenced as “event-to-sentence” in [2]) , which inspires us to use similar approaches to generate stories automatically.
> > > >
> > > > We are willing to further discuss with the reviewer if there are any remaining unaddressed concerns.

---

### Official Review · Reviewer_zk9y · 2023-10-31

**Soundness:** 2 fair
**Presentation:** 3 good
**Contribution:** 2 fair
**Rating:** 5
**Confidence:** 5

**Summary:**

This paper addresses the problem of automatic generation of story plots intended as a record of story premise, character descriptions, and sequence of plot elements. It purports to use LLMs for plot generation and considers limitations of state of the art systems such as DOC (Yang et al., 2022a)) in particular in their requirement for large number of calls to LLMs.
The work implements an end-to-end story plot generator, which replaces Open AI (as used in DOC) with Llama2-13B-chat and is fine-tunable with human feedback. The generator is based on on a two-level hierarchy (where DOC supports different numbers of levels for the hierarchical outline) and operates in a breadth-first and coarse-to-fine manner. The system has been evaluated using an evaluation prompt in GPT4, showing that (after using RLHF and SFT) it outperforms DOC in a majority of cases.

**Strengths:**

The paper addresses a difficult problem, which has a long history in AI and to which the advent of LLM offers new perspectives. It contains an appropriate rationale and demonstrate a reasonable knowledge of the state-of-the-art (to the exception of pre-2018 plan-based narrative generation). Experimental design is overall well described, leading to end-to-end model training.
While aiming at replicate and extend the performance of an existing approach (DOC) the work contains a number of original aspects (experimental approach, evaluation).
The paper also includes a transparent account of prompt engineering aspects faced during the development of the work, which might be beneficial to readers.

**Weaknesses:**

Although the paper describes generated units as 'plots' it stands in-between substantial previous work of Plan-based plot generation [Riedel and Young, 2010] where plot elements were narrative functions or operators and text-based story generation [Wang and Wan, 2019] (in the paper's references), originating in story completion experiments, up to systems to which the approach is compared that generate plot + narrative text [Yang et al., 2023 (2022a in the paper's references)]. It is thus unclear whether what is presented in the paper, in particular in Figure 7 are plots or storyboards, and this is not just a terminological issue, as it affects the ability to apply structural evaluation methods to plots (see below) as well as creating an unusual setting for users to 'evaluate' the plot as opposed to evaluation methods based on end-story quality or story understanding (e.g. QUEST graphs [Graesser et al., 1992] used in [Christian and Young, 2004]).
It would thus be necessary to much better justify the approach compared to end-story text generation (not just completion), or non-DL, non-LLM based plot generation (e.g. Plan-based).  In particular, considering that LLM text generation could be used in conjunction to other plot/backbone generation methods, or that the DOC method is in reality generating both plot and (textual) narrative.

Regarding reward models, there should probably be a discussion of previous approaches in text-based narrative generation, for instance [Ammanabrolu et al., 2019] and [Castricato et al., 2022]. There is also a lack of details on how RLHF has actually been performed (no details in the supplementary materials).

Evaluation techniques are somehow underspecified considering previous work in evaluating narrative generation. The expression of comparative preferences by GPT4 is moderately replicable and appears rather qualitative and not sufficiently related to structural properties of the plot and rigorous definitions of the above properties.
Although most of the work on evaluation based on narrative criteria (suspense, surprise, narrative arc...) has been developed as part of Plan-based narrative generation [Bae and Young, 2013] [Doust and Piwek, 2017] it should be transposable to DL-based (text-based [Yao et al., 2019] - in the paper's references, plot backbone [Polceanu et al., 2021]) or LLM-based work. Visual aspects of Plot structures that reconstruct Aristotelian arcs are of particular interest [Leong et al., 2022], not least because the paper makes reference [Goldfarb-Tarrant et al., 2020] to similar principles for neural-based story generation. In the absence of formal models it seems difficult to rely on GPT-4 with generic evaluation prompts, meaning that plots or storyboards would be better evaluated by industry professionals [Mirowski et al., 2022] (in the paper's references).
Other related work is not discussed [Xie et al., 2023], although it may have been released too late considering the ICLR deadline.

Ammanabrolu, P., Tien, E., Cheung, W., Luo, Z., Ma, W., Martin, L. and Riedl, M., 2019, August. Guided neural language generation for automated storytelling. In Proceedings of the Second Workshop on Storytelling (pp. 46-55).
Bae, B.C. and Young, R.M., 2013. A computational model of narrative generation for surprise arousal. IEEE Transactions on Computational Intelligence and AI in Games, 6(2), pp.131-143.
Castricato, L., Havrilla, A., Matiana, S., Pieler, M., Ye, A., Yang, I., Frazier, S. and Riedl, M., 2022. Robust Preference Learning for Storytelling via Contrastive Reinforcement Learning. arXiv preprint arXiv:2210.07792.
Christian, D.B. and Young, R.M., 2004, July. Comparing cognitive and computational models of narrative structure. In AAAI (pp. 385-390).
Doust, R. and Piwek, P., 2017, September. A model of suspense for narrative generation. In Proceedings of the 10th International Conference on Natural Language Generation (pp. 178-187).
Graesser, A.C., Gordon, S.E., Brainerd, L.E.: QUEST: a model of question answering. Comput. Math. Appl. 23, 733–745 (1992)
Leong, W., Porteous, J. and Thangarajah, J., 2022. Automated sifting of stories from simulated storyworlds. In: Proceedings of the Thirty-First International Joint Conference on Artificial Intelligence, IJCAI-22 (pp. 4950-4956).
Polceanu, M., Porteous, J., Lindsay, A. and Cavazza, M., 2021, May. Narrative plan generation with self-supervised learning. In Proceedings of the AAAI Conference on Artificial Intelligence (Vol. 35, No. 7, pp. 5984-5992).
Riedl, M.O., Young, R.M.: Narrative planning: balancing plot and character. J. Artif. Intell. Res. 39, 217–268 (2010)
Xie, Z., Cohn, T. and Lau, J.H., 2023, September. The Next Chapter: A Study of Large Language Models in Storytelling. In Proceedings of the 16th International Natural Language Generation Conference (pp. 323-351).

**Questions:**

How has RLHF been performed (user population, instructions, criteria...)?
How is the system dealing with relationships between characters? With the plot/character duality?
What is the average length of generated plots (counted in plot units or narrative functions)?

---

> ### Author Response · Authors · 2023-11-20
>
> We thank the reviewer for their helpful and insightful comments. Below we respond to the questions respectively.
>
> > Although the paper describes generated units as 'plots' it stands in-between substantial previous work of Plan-based plot generation… It is thus unclear whether what is presented in the paper, in particular in Figure 7 are plots or storyboards…
>
> We agree that the terminology ‘plot’ does not have a rigorous definition, and there are different choices of the format of the plot. The most suitable definition of the plot in our paper should be ‘some certain structure that captures most of the essential information of the story, which is often created in the planning stage’.
>
> In our work, we choose the structure of the plot (premise, settings, characters, outlines)  mainly due to the following reason:
>
> - The underlying principle is “coarse-to-fine”, i.e., to progressively provide more and more detailed information. This coarse-to-fine representation makes end2end autoregressive generation easier.
> - While short, this representation still contains a good amount of information about the story, which is helpful for multiple perspective evaluations (represented by multiple questions).
> - This representation is used by the SOTA story generation work DOC [1] for the planning stage. It can be used to generate high-quality full-length stories. This demonstrates that such a structure can lead to full story generation.
> - The plot structure we chose in this paper is text-based, which could greatly benefit from the powerful text-based LLMs that take the plot as input and produce the whole story (usually with multiple steps such as the second stage of DOC [1]).
>
>
> > Regarding reward models, there should probably be a discussion of previous approaches in text-based narrative generation, for instance [Ammanabrolu et al., 2019] and [Castricato et al., 2022].
>
> Compared to [Ammanabrolu et al., 2019],  where the reward is based on how close the generated event is to a pre-trained goal, our reward model training is more similar to CARP [Castricato et al., 2022], a contrastively-trained preference model as a reward signal in story generation. Note that our reward model is trained on story plots (using cross-entropy loss on the preference label) instead of the whole story, which makes learning and human labeling easier without losing important signals. We added these contents in Section 2.3 in the revision.
>
>
> > Evaluation techniques are somehow underspecified considering previous work in evaluating narrative generation...it seems difficult to rely on GPT-4 with generic evaluation prompts, meaning that plots or storyboards would be better evaluated by industry professionals.
>
> For GPT-4 evaluation, we provided the exact prompt format we used in Appendix B on Page 17. Also, thanks for your suggestion for conducting human evaluation. We conducted human evaluation during the rebuttal session and please see the global response for details. Given the short amount of rebuttal time, we won’t be able to find industry professionals to evaluate the story plots, but will open source all generated stories for future reference.
>
> > How has RLHF been performed (user population, instructions, criteria...)?
>
> After reward model training, we use standard RLHF objective (e.g., equation (2) in [2]):
>
> $\mathcal{L}\_\phi = \mathbb{E}\_{(x,y) \sim D\_{\pi\_\phi^{\text{RL}}}} [r\_\theta(x,y) - \beta \log( \pi\_\phi^{\text{RL}} (y|x) / \pi^{\text{SFT}} (y|x))]$,
>
> where $\pi\_\phi^{\text{RL}}$ is the learned policy by RLHF, $r\_\theta$ is the reward model, $\pi^{\text{SFT}}$ is the policy after SFT (our E2EPlot), and $\beta$ is to control the KL divergence. The prompt $x$ (i.e., the premise) is generated using the OpenPlot pipeline. We added the above formula in Section 2.3 in the revision.

---

> ### Author Response · Authors · 2023-11-20
>
> > How is the system dealing with relationships between characters? With the plot/character duality? What is the average length of generated plots (counted in plot units or narrative functions)?
>
> Our e2e model treats the story plot (premise, settings, characters, outlines) as a sequence and does not need to explicitly deal with the relationship between characters, which can be regarded as one of the advantages of the e2e story plot generator, compared to a hand-crafted method.
>
> For the OpenPlot pipeline (or original DOC [1]), after initializing the characters, when generating bulletin points of the outline, for each point, we will detect the characters that appeared in the sentence, and refine the character portraits by adding details in the current sentence to the initial character portrait.
>
> On the other hand, we acknowledge that such a modeling relies on LLM’s capacity to model characters and their relationships/interactions, and in many situations, still yields a shallow and boring plot. In the future, we will think about ways to model their relationships in more detail, not by hand-crafting but by leveraging the power of LLMs.
>
> The average length of the generated plots is ~1000 tokens.
>
> **References:**
>
> [1] Yang, Kevin, Dan Klein, Nanyun Peng, and Yuandong Tian. "Doc: Improving long story coherence with detailed outline control." arXiv preprint arXiv:2212.10077 (2022).
>
> [2] Ouyang, Long, Jeffrey Wu, Xu Jiang, Diogo Almeida, Carroll Wainwright, Pamela Mishkin, Chong Zhang et al. "Training language models to follow instructions with human feedback." Advances in Neural Information Processing Systems 35 (2022): 27730-27744.

---

> > ### Comment · Reviewer_zk9y · 2023-11-22
> > **Acknowledgement of Authors Response**
> >
> > Thanks for your response and clarification, in particular on the reward model, which was appropriate.
> > I am however less convinced by other elements of response, in particular on the definition of plot which leads to a specific form of narrative generation difficult to situate with respect to previous work. Similar comments could be made on evaluation techniques, especially for high-level narrative phenomena such as suspense or surprise. I am not implying that your assumptions are technically flawed, but instead that, by being somehow at variance with previous work (DL-based or not) on story generation, they might end up addressing problems of lesser relevance.
> > For this reason, I am not inclined to modify my assessment.

---

> > > ### Author Response · Authors · 2023-11-23
> > >
> > > We thank the reviewer for reading our response and providing further feedback.
> > >
> > > **[Defintion of plots and difference from previous literature]** As we mentioned in the previous response, the (premise, setting, characters, outlines) structure of our plot is proven (by the SOTA story generation work DOC [1])  to be efficient in carrying the essential information of the whole story and generating the whole story. We agree that our plot has a different format from those of some of the previous work, but it is hard or even impossible and maybe unnecessary to propose an efficient format of plot that can unify all the previous formats of plots. We also want to emphasize that our work is an LLM-based story generation method, which makes it natural to choose a text-based plot that might be different from previous work that is not LLM-based.  The LLM-based nature of our work makes our story plot different from some previous non-LLM-based works, and this nature makes it possible to generate a story plot at a much faster speed, and our generator model can be easily improved from feedback.
> > >
> > > **[Evaluation techniques]** Comparison-based evaluation technique on a specific aspect such as interestingness or the overall quality is also used and proven to be efficient by previous works such as DOC [1]. We emphasize that these aspects are good evaluation metrics on story plots because:
> > > 1. These high-level aspects remain roughly invariant on the whole story and the story plot;
> > > 2. The plots are text-based and relatively short, which can either be read by humans or parsed by LLMs easily.
> > >
> > > We are happy to address any further questions.

---

### Author Response · Authors · 2023-11-20
**Global response (1/3)**

We thank all the reviewers for their efforts in reviewing our paper and providing helpful and insightful feedback. Below we first respond to some common questions.

**[Name of different models in the revision and rebuttal]**
To make the presentation more clear and concise, we name the rebuilt pipeline from the original DOC **OpenPlot**, name the end-to-end model after SFT **E2EPlot**, and name the model after RLHF **RLPlot**. Note that we have five different **RLPlot** models w.r.t. five different aspects, and we will also use **RLPlot_Q1, …,  RLPlot_Q6** to refer to specific RLPlot models w.r.t. corresponding aspects.

**[Novelty]**
While we admit that the techniques used in this paper are quite standard, we are targeting “a difficult problem”, as suggested by the reviewer zk9y. To put this comment into full context, reviewer zk9y commented that our paper “addresses a difficult problem, which has a long history in AI and to which the advent of LLM offers new perspectives”, Also, our work “contains a number of original aspects (experimental approach, evaluation)” and “includes a transparent account of prompt engineering aspects faced during the development of the work, which might be beneficial to readers”. Therefore, from our point of view, using standard approaches/pipelines should not be regarded as a lack of novelty.

If we agree that solving such a problem is valuable, then our work indeed leads to strong performance gain over existing methods like DOC (story plot stage), e.g., 10x faster generation of story plot with comparable performance verified by both GPT4 and human evaluation, with much fewer LLM calls (and lower cost given that LLaMA-2 can be served locally).

**[Why story plot rather than the entire story]**
Story plots (~1000 tokens) are often much shorter than the entire story (often more than 5000 tokens, or even longer) and yet capture the essence of the story. Compared to the full ones, this makes the story plot much easier to evaluate for both LLMs and human evaluators, and still the evaluation is well-defined (or grounded) to the actual points. This also makes any story-plot based generator relatively easy to fine-tune.

**[Human evaluation results]**
We added human evaluation results using the prolific platform. Overall the results are quite consistent with GPT4.

For E2E plot versus OpenPlot, humans also agree that their outputs are comparable in quality, which is consistent with GPT4’s evaluation.

For E2E plot versus RLPlot, we also see strong consistency between GPT4 results and human evaluations:

  - According to GPT4 evaluation, our RLHF fine-tuned model performs better in Q1, Q4, Q5, Q6, and worse in Q3.

  - According to human evaluation, our RLHF fine-tuned model performs better in Q1, Q4, Q5, and worse in Q3 and Q6.

Therefore, there is only one discrepancy (Q6). Note that Q6 is “Which story has a better ending?” and thus can be very subjective. For example, some reviewers may like a happy ending while others may like a sad ending. This is pointed out also in other concurrent works regarding story evaluation [1].

**References:**

[1] Learning Personalized Story Evaluation, https://aps.arxiv.org/abs/2310.03304.

---

> ### Author Response · Authors · 2023-11-20
> **Global response (2/3)**
>
> Finally, we present the detailed result of human evaluation. Since we have five RLPlot models, we name them RLPlot_Q1, RLPlot_Q3, RLPlot_Q4, RLPlot_Q5, and RLPlot_Q6, respectively.
> We sent 500 (OpenPlot vs E2EPlot)  + 300 * 5 (RLPlot vs E2EPlot) = 2000 story plot pairs, which are the same as what we used for GPT4 evaluation, to prolific, and ~1700 of them are labeled (each pair requires 1 label, each participant are required to label five pairs, and some of the participants labeled part of the five pairs). For each pair, the participants are required to answer seven questions, where Q1-Q6 are the same as in Table 2 in our paper, Q2 is a free text explanation for Q1, and Q7 is ‘Which story is better in overall quality’.
>
> Note that in our original paper, for each story pair, we only asked GPT4 to evaluate one question. For OpenPlot vs E2EPlot, the question is overall quality; for RLPlot vs E2EPlot, the question is the corresponding aspect. For our human evaluation result, we provide results of all six (5+1) questions for each label. For completeness, we also conducted GPT4-evaluation for each of the 2000 pairs on all six questions. Below are the results of the human evaluation. (which can also be found in our revision on page 18-19, table 8-13, appendix C.1)
>
>
>
> **OpenPlot vs E2EPlot  (sent 500, labeled 410)**
>
> |     | OpenPlot wins | E2EPlot wins |    Tie    |
> |:---:|:-------------:|:------------:|:---------:|
> | Q1  |     41.2%     |     43.4%    |   15.4%   |
> | Q3  |     40.5%     |     40.5%    |   19.0%   |
> | Q4  |     43.2%     |     40.7%    |   16.1%   |
> | Q5  |     40.2%     |     39.5%    |   20.2%   |
> | Q6  |     39.0%     |     39.8%    |   21.2%   |
> | Q7  |     38.5%     |     41.2%    |   20.2%   |
>
>
>
>
>
>
> **RLPlot_Q1 vs E2EPlot  (sent 300, labeled 248)**
>
> |     | RLPlot_Q1 wins | E2EPlot wins |    Tie    |
> |:---:|:--------------:|:------------:|:---------:|
> | Q1  |     48.8%      |    37.9%     |   13.3%   |
> | Q3  |     44.8%      |    38.3%     |   16.9%   |
> | Q4  |     46.8%      |    32.3%     |   21.0%   |
> | Q5  |     40.7%      |    35.1%     |   24.2%   |
> | Q6  |     47.6%      |    35.1%     |   17.3%   |
> | Q7  |     43.1%      |    36.3%     |   20.6%   |
>
>
>
>
> **RLPlot_Q3 vs E2EPlot  (sent 300, labeled 241)**
>
>
> |     | RLPlot_Q3 wins | E2EPlot wins |    Tie    |
> |:---:|:--------------:|:------------:|:---------:|
> | Q1  |     41.5%      |    41.1%     |   17.4%   |
> | Q3  |     38.2%      |    41.1%     |   20.7%   |
> | Q4  |     34.4%      |    45.6%     |   19.9%   |
> | Q5  |     35.3%      |    38.6%     |   26.1%   |
> | Q6  |     39.4%      |    38.2%     |   22.4%   |
> | Q7  |     36.1%      |    41.9%     |   22.0%   |
>
>
>
> **RLPlot_Q4 vs E2EPlot  (sent 300, labeled 252)**
>
> |     | RLPlot_Q4 wins | E2EPlot wins |    Tie    |
> |:---:|:--------------:|:------------:|:---------:|
> | Q1  |     41.3%      |    40.9%     |   17.9%   |
> | Q3  |     42.5%      |    39.3%     |   18.3%   |
> | Q4  |     40.1%      |    38.5%     |   21.4%   |
> | Q5  |     37.7%      |    38.5%     |   23.8%   |
> | Q6  |     40.1%      |    38.9%     |   21.0%   |
> | Q7  |     40.1%      |    38.1%     |   21.8%   |
>
>
>
>
> **RLPlot_Q5 vs E2EPlot  (sent 300, labeled 251)**
>
>
> |     | RLPlot_Q5 wins | E2EPlot wins |    Tie    |
> |:---:|:--------------:|:------------:|:---------:|
> | Q1  |     43.4%      |    43.0%     |   13.5%   |
> | Q3  |     45.0%      |    42.6%     |   12.4%   |
> | Q4  |     46.2%      |    42.6%     |   11.2%   |
> | Q5  |     43.8%      |    37.5%     |   18.7%   |
> | Q6  |     41.8%      |    38.2%     |   19.9%   |
> | Q7  |     43.0%      |    37.8%     |   19.1%   |
>
>
>
> **RLPlot_Q6 vs E2EPlot  (sent 300, labeled 248)**
>
> |     | RLPlot_Q6 wins | E2EPlot wins |    Tie    |
> |:---:|:--------------:|:------------:|:---------:|
> | Q1  |     38.7%      |    44.0%     |   17.3%   |
> | Q3  |     35.9%      |    44.8%     |   19.4%   |
> | Q4  |     39.5%      |    41.1%     |   19.4%   |
> | Q5  |     35.1%      |    39.1%     |   25.8%   |
> | Q6  |     36.3%      |    45.2%     |   18.5%   |
> | Q7  |     35.9%      |    41.1%     |   23.0%   |

---

> > ### Author Response · Authors · 2023-11-20
> > **Global response (3/3)**
> >
> > Now, we also present the GPT-4 evaluation results for each pair on all six questions (which can also be found in our revision on Appendix C.2, Table 14-19) :
> >
> > **OpenPlot vs E2EPlot  (GPT4 evaluation, total 500)**
> >
> > |     | OpenPlot wins | E2EPlot wins |    Tie    |
> > |:---:|:-------------:|:------------:|:---------:|
> > | Q1  |     53.2%     |     45.2%    |     1.6%   |
> > | Q3  |     53.6%     |     40.6%    |     5.8%   |
> > | Q4  |     53.2%     |     45.6%    |     1.2%   |
> > | Q5  |     53.0%     |     45.2%    |     1.8%   |
> > | Q6  |     48.2%     |     50.4%    |     1.4%   |
> > | Q7  |     45.8%     |     46.8%    |     7.4%   |
> >
> > **RLPlot_Q1 vs E2EPlot  (GPT4 evaluation, total 300)**
> >
> > |     | RLPlot_Q1 wins | E2EPlot wins |    Tie    |
> > |:---:|:-------------:|:------------:|:---------:|
> > | Q1  |     54.0%     |     44.0%    |     1.6%   |
> > | Q3  |     53.0%     |     41.0%    |     6.0%   |
> > | Q4  |     49.0%     |     50.7%    |     0.3%   |
> > | Q5  |     51.7%     |     46.7%    |     1.7%   |
> > | Q6  |     56.7%     |     41.0%    |     2.3%   |
> > | Q7  |     49.7%     |     44.3%    |     6.0%   |
> >
> >
> > **RLPlot_Q3 vs E2EPlot  (GPT4 evaluation, total 300)**
> >
> > |     | RLPlot_Q3 wins | E2EPlot wins |    Tie    |
> > |:---:|:-------------:|:------------:|:---------:|
> > | Q1  |     45.3%     |     52.0%    |     2.7%   |
> > | Q3  |     46.0%     |     50.3%    |     3.7%   |
> > | Q4  |     43.3%     |     56.3%    |     0.3%   |
> > | Q5  |     44.0%     |     55.0%    |     1.0%   |
> > | Q6  |     49.7%     |     47.3%    |     3.0%   |
> > | Q7  |     46.7%     |     44.3%    |     9.0%   |
> >
> >
> > **RLPlot_Q4 vs E2EPlot  (GPT4 evaluation, total 300)**
> >
> > |     | RLPlot_Q4 wins | E2EPlot wins |    Tie    |
> > |:---:|:-------------:|:------------:|:---------:|
> > | Q1  |     55.0%     |     43.0%    |     2.0%   |
> > | Q3  |     51.7%     |     42.0%    |     6.3%   |
> > | Q4  |     60.0%     |     39.3%    |     0.7%   |
> > | Q5  |     51.7%     |     45.7%    |     2.7%   |
> > | Q6  |     58.7%     |     40.3%    |     1.0%   |
> > | Q7  |     44.7%     |     46.0%    |     9.3%   |
> >
> > **RLPlot_Q5 vs E2EPlot  (GPT4 evaluation, total 300)**
> >
> > |     | RLPlot_Q5 wins | E2EPlot wins |    Tie    |
> > |:---:|:-------------:|:------------:|:---------:|
> > | Q1  |     40.3%     |     56.0%    |     3.7%   |
> > | Q3  |     42.0%     |     53.0%    |     5.0%   |
> > | Q4  |     41.7%     |     57.0%    |     1.3%   |
> > | Q5  |     50.0%     |     48.0%    |     2.0%   |
> > | Q6  |     51.0%     |     47.3%    |     1.7%   |
> > | Q7  |     43.3%     |     49.7%    |     7.0%   |
> >
> >
> >
> >
> > **RLPlot_Q6 vs E2EPlot  (GPT4 evaluation, total 300)**
> >
> > |     | RLPlot_Q6 wins | E2EPlot wins |    Tie    |
> > |:---:|:-------------:|:------------:|:---------:|
> > | Q1  |     44.3%     |     53.7%    |     2.0%   |
> > | Q3  |     42.7%     |     53.7%    |     3.7%   |
> > | Q4  |     44.7%     |     55.0%    |     0.3%   |
> > | Q5  |     42.3%     |     54.3%    |     3.3%   |
> > | Q6  |     53.7%     |     42.3%    |     4.0%   |
> > | Q7  |     44.7%     |     50.0%    |     5.3%   |

---

### Meta-Review · Area_Chair_rufr · 2023-12-06

**Metareview:**

The author's rebuttal has made efforts to address several concerns, particularly the human evaluation, which has been quite helpful.

However, even after the author's rebuttal, a major concern remains centered on the novelty of this work. The specific discussions regarding novelty can be summarized as follows:

1. Given the existence of the DOC work, it appears that the paper offers limited new insights into plot-based story generation.
2. The authors argue that tackling a challenging problem adds to the novelty of their work. However, there is a debate as to whether merely addressing a difficult problem is sufficient.
3. There is also a debate about whether "speedup" can be considered a substantial contribution and novelty in this context.

The author responses have not effectively addressed Point 1.

Regarding Point 2, there is partial agreement. While it is recognized that targeting a challenging problem is commendable, the typical approach in our field involves formally defining the problem and creating new benchmarks. (We understand that the aforementioned process is not easy for this research direction). However, given the ongoing inconsistency regarding the definition of plots, it is argued that the paper may not fully meet the standards for targeting a difficult problem as a novelty.

For Point 3, it is acknowledged that some reviewers agree that speed improvement can be considered a contribution. However, one of the reviewer also raised a valid point regarding which is a more hardcore challenge in this direction.

Due to the remaining unanswered questions and concerns, both the (updated) submission and the author responses have not managed to fully convince all reviewers of the paper's value.

**Justification For Why Not Higher Score:**

The questions 1 and 2 about the novelty and contributions remain unaddressed by the author responses, which is important for the reviewers to convince the value of the submission.

**Justification For Why Not Lower Score:**

N/A -- this is a rejection

---

### Decision · Program_Chairs · 2024-01-16

Reject